# Accuracy of Personalized Computed Tomographic 3D Templating for Acetabular Cup Placement in Revision Arthroplasty

**DOI:** 10.3390/medicina59091608

**Published:** 2023-09-06

**Authors:** Philipp Winter, Ekkehard Fritsch, Thomas Tschernig, Lars Goebel, Milan Wolf, Manuel Müller, Julius J. Weise, Patrick Orth, Stefan Landgraeber

**Affiliations:** 1Department of Orthopaedic Surgery, University of Saarland, Kirrberger Straße, 66421 Homburg, Germany; ekkehard.fritsch@uks.eu (E.F.); lars.goebel@uks.eu (L.G.); milan.wolf@uks.eu (M.W.); manuelmueller@gmx.net (M.M.); patrick.orth@uks.eu (P.O.); stefan.landgraeber@uks.eu (S.L.); 2Institute of Anatomy, University of Saarland, Kirrberger Straße, 66421 Homburg, Germany; thomas.tschernig@uks.eu; 3Department of Medical Biometry, Epidemiology and Medical Informatics, University of Saarland, Kirrberger Straße, 66421 Homburg, Germany; juw@med-imbei.uni-saarland.de

**Keywords:** revision arthroplasty, hip (joint), acetabular cup, 2D, 3D, computed tomography

## Abstract

*Background*: Revision hip arthroplasty presents a surgical challenge, necessitating meticulous preoperative planning to avert complications like periprosthetic fractures and aseptic loosening. Historically, assessment of the accuracy of three-dimensional (3D) versus two-dimensional (2D) templating has focused exclusively on primary hip arthroplasty. *Materials and Methods*: In this retrospective study, we examined the accuracy of 3D templating for acetabular revision cups in 30 patients who underwent revision hip arthroplasty. Utilizing computed tomography scans of the patients’ pelvis and 3D templates of the implants (Aesculap Plasmafit, B. Braun; Aesculap Plasmafit Revision, B. Braun; Avantage Acetabular System, Zimmerbiomet, EcoFit 2M, Implantcast; Tritanium Revision, Stryker), we performed 3D templating and positioned the acetabular cup implants accordingly. To evaluate accuracy, we compared the planned sizes of the acetabular cups in 2D and 3D with the sizes implanted during surgery. *Results*: An analysis was performed to examine potential influences on templating accuracy, specifically considering factors such as gender and body mass index (BMI). Significant statistical differences (*p* < 0.001) in the accuracy of size prediction were observed between 3D and 2D templating. Personalized 3D templating exhibited an accuracy rate of 66.7% for the correct prediction of the size of the acetabular cup, while 2D templating achieved an exact size prediction in only 26.7% of cases. There were no statistically significant differences between the 2D and 3D templating methods regarding gender or BMI. *Conclusion*: This study demonstrates that 3D templating improves the accuracy of predicting acetabular cup sizes in revision arthroplasty when compared to 2D templating. However, it should be noted that the predicted implant size generated through 3D templating tended to overestimate the implanted implant size by an average of 1.3 sizes.

## 1. Introduction

The growing prevalence of primary arthroplasty of the hip is expected to lead to an increase in the number of revision surgeries in the future [1,2,3]. Based on projections, the incidence of revision hip arthroplasty will rise by as much as 43% to 70% between the years 2014 and 2030 [2]. The indications for revision total hip arthroplasty encompass a range of factors with common causes, including aseptic loosening (52%), instability (17%), infection (5.5%), debilitating pain, periprosthetic fractures, or component failure [4].

In a recently published study, it was demonstrated that scaling discrepancies in anteroposterior pelvic radiographs used for templating in total hip arthroplasty planning can result in errors of ≥3 mm in 25% of patients and ≥6 mm in 2% of patients, when compared to CT-based planning [5]. CT scans are also used for biomechanical analysis and modeling of hip implants, allowing simulation of material combinations and thus implant improvement [6,7,8,9].

Revision hip arthroplasty poses significant challenges, particularly in cases involving severe bone loss, and necessitates meticulous preoperative planning to achieve precise intraoperative outcomes. Preoperative planning for primary hip arthroplasty has long been recognized as a means to reduce complications such as periprosthetic fracture or aseptic loosening resulting from inaccurate estimation of implant size [4,10,11,12,13,14]. Additionally, it has resulted in other benefits, such as decreased surgical time, improved postoperative stability, and greater range of motion. Furthermore, preoperative planning has a growing legal significance, enabling surgeons to demonstrate their thoroughness and potentially mitigating allegations of negligence [15,16,17]. Postoperative leg length discrepancy (LLD) following total hip replacement is a common issue, and it is the most frequent cause of legal disputes involving orthopedic surgeons [18,19,20,21]. LLD exceeding ±5 mm is typically noticeable to the patient, and it can lead to discomfort, potentially requiring further revision surgeries. Therefore, particularly in revision hip arthroplasty, meticulous preoperative planning is necessary in order to address this concern [22,23]. Studies investigating ways of enhancing preoperative planning have compared three-dimensional (3D) templating with two-dimensional (2D) templating in primary hip arthroplasty, and the results have shown that 3D templating provides greater accuracy and reliability in predicting implant size [24,25,26]. In a systematic review by Bishi et al., it was demonstrated that the 3D template provides higher accuracy and reliability in predicting implant size in preoperative planning compared with the 2D template in primary arthroplasty [25]. However, in both clinical practice and the current literature, 2D templating still endures as the standard technique [27,28]. Existing studies on preoperative planning in revision arthroplasty have predominantly centered around the application of 3D printing [29,30]. 

To date, there are no studies available regarding the accuracy of 3D templating in revision hip arthroplasty. The objective of this study is to assess the precision in acetabular cup positioning using computed tomographic (CT) 3D templating in revision arthroplasty compared to conventional 2D templating.

## 2. Materials and Methods

This retrospective study included all patients who underwent revision arthroplasty with revision of the acetabular cup at our institution, performed by senior surgeons, between April 2019 and November 2022. Inclusion criteria were all patients who underwent revision hip arthroplasty in our institution with reimplantation of an acetabular cup between April 2019 and November 2022 and who had routine preoperative computed tomography of the pelvis. Patients who did not undergo acetabular cup revision or who did not have preoperative computed tomography were excluded. The included implants were: Aesculap Plasmafit, B. Braun, Melsungen, Germany; Aesculap Plasmafit Plus 3, B. Braun, Melsungen, Germany; Aesculap Plasmafit Plus 7, B. Braun, Melsungen, Germany; Aesculap Plasmafit Revision, B. Braun, Melsungen, Germany; Avantage Acetabular System, Zimmer Biomet, Warsaw, IN, USA; EcoFit 2M, Implantcast, Buxtehude, Germany; Tritanium Revision, Stryker, Mahwah, NJ, USA.

The study respected the ethical standards for biomedical research in accordance with the Declaration of Helsinki [31]. It received approval from the local Ethics Committee (reference number 160/22) and complied with the ethical review requirements of our institution.

### 2.1. Preoperative Imaging

All patients included in the study underwent standard conventional radiography as part of their diagnostic evaluation. This involved anteroposterior (A/P) views of the pelvis and lateral views of the hip in the Lauenstein position. Additionally, preoperative CT scans of the pelvis were routinely conducted to assess the extent of bony defects and aid in surgical planning. A helical CT scanner with a slice thickness of 0.75 mm was utilized for image acquisition. The acquired CT data were subsequently transferred to our Picture Archiving and Communication System (PACS), Sectra, Sweden (Sectra AB, Linköping, Sweden).

### 2.2. Digital Templating

Prior to surgery, conventional 2D digital templating was conducted using X-ray images in two planes. Two-dimensional planning is performed with implant templates based on digital X-ray images. To determine the magnification factor, a radiopaque metal ball with a standardized diameter of 25.0 mm was placed between the patient’s legs at the level of the hip joint rotation center. Preoperative 2D planning was performed by senior surgeons using the Sectra 2D planning system (Sectra AB, Linköping, Sweden) and the corresponding 2D templates for the acetabular cup.

The 3D modeling and templating process for the acetabular cup was based on the analysis of CT imaging of the pelvis, as depicted in Figure 1. The 3D planning is performed in three planes (axial, saggital and coronary). For this process, the Sectra Joint Replacement Tool (“3D Joint Sectra”), a medical imaging IT solution from Sectra (Sectra AB, Linköping, Sweden), was employed. This tool utilized the 3D templates for the acetabular cup (Aesculap Plasmafit, B. Braun, Melsungen, Germany; Aesculap Plasmafit Plus 3, B. Braun, Melsungen, Germany; Aesculap Plasmafit Plus 7, B. Braun, Melsungen, Germany; Aesculap Plasmafit Revision, B. Braun, Melsungen, Germany; Avantage Acetabular System, Zimmer Biomet, Warsaw, IN, USA; EcoFit 2M, Implantcast, Buxtehude, Germany; Tritanium Revision, Stryker, Mahwah, NJ, USA). The 3D templating was performed independently of the previous 2D templating, and it was compared with the actual size of the implanted acetabular cup by an independent examiner, who then evaluated the accuracy of the 3D templating in predicting the appropriate implant size.

### 2.3. Surgical Procedure

All patients included in the study underwent surgery using a lateral approach. The surgery was performed by one of three senior surgeons. The specific types of acetabular cups used are outlined in Table 1. The surgical objectives focused on achieving stable anchoring of the acetabular cup within the host bone. Additionally, the surgical procedures aimed to restore natural biomechanical conditions, including leg length, center of rotation, and lateralization.

Cementless implantation of the acetabular cup was performed in 27 cases. In three cases, a cemented implantation was performed. In one case, an acetabular wedge augment was used to achieve adequate stability of the implant (Figure 2). To ensure optimal positioning, the acetabular cup was placed within the Lewinnek safety zone, with an inclination range of 40° ± 10° and an anteversion range of 15° ± 10° [32]. These ranges were selected to maintain the desired biomechanical stability and function of the hip joint.

### 2.4. Statistical Analysis

Statistical analyses were performed using the SPSS software package (Version 29; IBM SPSS Statistics, Chicago, IL, USA). A two-sided significance level of *p* < 0.05 was used to determine statistical significance. To assess systematic differences between 2D and 3D templating and the implant size used during surgery, contingency tables and the Sign tests were employed. The contingency tables allowed for the comparison of categorical variables, while the Sign tests were used for non-parametric paired comparisons. The accuracy of size predictions by the two planning methods, referred to as accuracy, was calculated and compared between 2D and 3D templating using McNemar’s test, which is suitable for analyzing matched-pair data. To examine the influence of the patient’s sex on the accuracy of templating, Fisher’s test for 2 × 2 contingency tables was utilized. This test allowed for the assessment of associations between categorical variables. The influence of body mass index (BMI) on planning accuracy was examined using the Mann–Whitney U test. This test was chosen for comparing patient groups with correctly predicted sizes to those with incorrectly predicted sizes, taking into account the non-parametric nature of BMI data.

## 3. Results

Strictly adhering to the predetermined inclusion and exclusion criteria, a cohort of 30 cases was selected for the present study. Within this cohort, 46% (*n* = 14) of the patients were female and 53% (*n* = 16) were male. The mean age of the entire cohort was 71 years (±9.6), the female patients had a mean age of 70.5 years (±11.7), and the male patients had a mean age of 72.4 years (±7.5). The demographic profiles of the participants are presented in Table 2. In accordance with the World Health Organization’s criteria [22], five patients were classified as within the normal weight range, while fifteen patients were classified as overweight, and ten were classified as obese. The process of 2D preoperative templating was executed by three highly skilled senior surgeons, whereas the personalized 3D templating procedure was carried out by a resident under the supervision and guidance of the senior surgeons. A total of 13 patients underwent surgery for aseptic loosening and 17 patients for infection. 

The precise determination of the acetabular cup size during intraoperative assessment showed that personalized 3D templating yielded accurate predictions for 20 of the 30 patients (66.7%), while conventional 2D templating achieved this level of accuracy for only 8 of the 30 patients (26.7%), (McNemar test, *p* < 0.002), as presented in Table 3. Notably, 2D templating demonstrated a higher frequency of overestimation rather than underestimation of the intraoperatively determined size, though the difference was not statistically significant (Sign test, *p* = 0.523). The size was over- or underestimated by one size in 12 cases (12/30). A discrepancy of 2 implant sizes was observed in 5 cases (5/30), and in 5 other cases, the prediction deviated by 3 implant sizes (5/30). The calculated mean difference of correct acetabular size prediction in 2D templating was 1.68, with a standard deviation of 0.93 and a range discrepancy of −3 to +3 sizes between the template size and the size finally implanted.

Regarding 3D templating, the intraoperatively determined size was underestimated in one case, while overestimations were observed in 9 cases (Sign test, *p* = 0.109). In 7 cases, the correct size was under- or overestimated by one size (7/30), while 3 cases showed a discrepancy of 2 implant sizes (3/30). The calculated mean difference of accurate acetabular cup size prediction in the context of 3D templating was determined to be 1.3, accompanied by a standard deviation of 0.67. Furthermore, a range discrepancy of −2 to +2 sizes was observed between the template size and the size finally implanted.

The acetabular cup alignment was within the Lewinnek safe zone in 93.3% (28/30) and had a mean inclination angle of 43.9°. 

### 3.1. Gender and Planning Accuracy

Conventional 2D templating accurately predicted the correct implant size in only 21.4% of female patients (*n* = 3/14) and 31.3% of male patients (*n* = 5/16), as determined through Fisher’s exact test (*p* = 0.689). In contrast, 3D templating demonstrated a higher rate of correct prediction of implant size, achieving accurate results for 81.3% of the male patients (*n* = 13/16) and 50% of the female patients (*n* = 7/14). Statistical analysis using Fisher’s test indicated no significant difference between genders in terms of the accuracy of 3D templating (*p* = 0.122), as shown in Table 3.

### 3.2. BMI and Planning Accuracy

According to the criteria outlined by the World Health Organization (WHO) [33], the patients were classified into four groups based on their body mass index (BMI): underweight, normal weight, overweight, and obese (Table 2). Using 3D templating, the accurate size of the acetabular cup was successfully predicted in 3 out of 5 patients classified as normal weight, 11 out of 15 patients classified as overweight, and 6 out of 10 patients classified as obese. In contrast, using 2D templating, the accurate size was predicted in 0 out of 5 patients classified as normal weight, 7 out of 15 patients classified as overweight, and 1 out of 10 patients classified as obese. Statistical analysis using the Mann–Whitney U test indicated no statistically significant difference in BMI between the correctly and incorrectly predicted sizes for both 2D templating and 3D templating (*p* = 0.765 and *p* = 0.812, respectively), as shown in Table 3.

## 4. Discussion

The main finding of our study is that 3D templating based on computed tomography leads to an increased accuracy of predicting the size of acetabular cups in revision arthroplasty and therefore provides a reliable tool for preoperative planning in revision hip arthroplasty. Regarding the precision of planning in revision arthroplasty, the current literature is still limited. Existing studies on preoperative planning in revision arthroplasty have predominantly centered around the application of 3D printing [29,30]. In the study by Hughes et al., life-size 3D models were manufactured for three complex cases of acetabular revision hip arthroplasty from the CT scans. The surgeon’s preoperative planning with the 3D model resulted in increased surgical precision and a simultaneous reduction in complications. One advantage of life-size 3D models in revision hip arthroplasty is certainly the surgeon’s tactile assessment of acetabular bone defects [29]. However, the tactile sensations that can be achieved through 3D printing cannot be replicated in digital simulations. Nonetheless, future advances may include the use of optical, see-through devices that project a holographic rendering of the 3D model onto the surgical site, allowing simulation of accurate sizing and positioning. Although initial experimental studies have been performed in this area, they are limited to primary arthroplasty scenarios [34,35].

The prediction of implant sizes in revision hip arthroplasty is particularly challenging due to the acetabular defect situation and the presence of endoprostheses. Precise templating based on X-rays and imaging modalities, such as CT scans, is a crucial element in the preoperative planning of acetabular cup revision. The primary objective is to achieve an accurate prediction of implant size and optimal positioning and thus enhance the efficiency and outcome of the surgical procedure. Computed tomography has been widely adopted in medical investigations [36,37].

Kearney et al. demonstrated a 27% accuracy in acetabular cup size prediction in primary hip arthroplasty through acetate templating (14/51). Schiffner et al. were able to accurately predict the size of the acetabular cup by 2D templating in 44.8% of cases and in 58.6% by 3D templating.

Furthermore, in a systematic review conducted by Bishi et al., the reported range for exact acetabular cup size prediction was between 25% and 85.7% [25]. Using 3D templating in primary hip arthroplasty, the range of exact size prediction increases significantly from 40% to 98% [25]. These findings are consistent with the results obtained in our study regarding revision hip arthroplasty. We observed an accuracy of preoperative 3D templating of 66.7 % (20/30) in comparison with 26.7% (8/30) by preoperative 2D templating. 

Considering the tolerance range encompassing one size above or below the definitive implant size (±1), Kearney et al. demonstrated a greater accuracy of 75% (38/51), accompanied by a calculated mean difference of 1.29 sizes and a standard deviation of 2.56 [38]. In the study by Schiffner et al., an accuracy of 83.6% in 2D templating and of 86.2% in 3D templating was achieved for the prediction of acetabular cup size (±1). These findings align with the results obtained in our study. In 90% (27/30), we were able to predict the size accurately (+/− 1 size) through 3D templating. 

Kearney et al. described a range discrepancy of −2 to +4 [38]. In our study with 3D templating for revision hip arthroplasty, we achieved a higher accuracy level ranging from −2 to +2. Nevertheless, these findings show that, on average, the template size determined through 3D templating for the acetabulum exhibited an overestimation of 1.3 sizes with a standard deviation of 0.67 from the implanted cup size. However, we observed a calculated mean difference of 1.68 sizes with a standard deviation of 0.93 for 2D templating.

Despite the fact that the radiation exposure associated with 3D templating using CT scans is approximately five times higher than that of 2D templating using radiographs, the dose of 11–12 mSv utilized in the former is well below the reported threshold for an increased risk of cancer (200–5000 mSv) [24,39,40]. In the context of revision arthroplasty, pelvic CT is commonly conducted as part of the standard protocol to provide a comprehensive evaluation of the acetabular bone condition. Consequently, in the specific context of revision arthroplasty, where CT scans are routinely performed, the utilization of CT does not result in an elevated radiation exposure beyond what is already customary and expected. Furthermore, Kaiser et al. reported the utilization of a novel tin-filtered ultra-low-dose CT technique for the hip, resulting in a radiation dose of 0.58 mSv, which is comparable to conventional radiographs [5].

Regarding BMI, the impact of the accuracy of both 2D and 3D templating did not yield statistically significant findings. These results align with the studies conducted by Holzer et al. [41]. In Holzer et al.’s research, no significant correlation was observed between BMI and the accuracy of acetabular templating. However, a significant difference was noted in femoral component planning accuracy between patients of normal weight and those who were overweight [41]. Furthermore, no significant disparities were observed in our study between the planning accuracy of both 2D and 3D templating with regards to gender. Therefore, the reliability of preoperative 3D templating can be presumed to be consistent for both male and female patients. These findings are again in agreement with the study conducted by Holzer et al., which also indicated the absence of statistically significant gender-specific differences in planning accuracy and calibration deviations of X-rays between men and women [41].

It is important to acknowledge certain limitations of our study, including the absence of calculations for intra- and interobserver reliability. Second, we studied only a small cohort of patients. Additionally, our investigation focused on seven acetabular cup systems. A wide range of other revision acetabular cups are available on the market, which were not considered in this study. Although the increased accuracy observed with 3D templating is promising, further analysis is needed to determine the potential impact on clinical outcomes.

## 5. Conclusions

This study shows an increased accuracy of predicting the size of acetabular cups in revision hip arthroplasty by using 3D templating compared to 2D templating. Nevertheless, the implant size determined through 3D templating for the acetabulum exhibited an overestimation of 1.3 sizes. Until now, evaluation of the accuracy of 3D and 2D templating has primarily focused on primary arthroplasty, while the potential of personalized 3D planning in the context of revision arthroplasty remains largely unexplored. An additional advantage of 3D templating is that it allows effective preoperative planning of acetabular wedge augments, thereby facilitating intraoperative positioning and providing improved implant stability. Further prospective studies are expected to provide a deeper understanding on the crucial aspects of patients’ clinical outcome and a reduction in surgical time by using personalized 3D templating in revision arthroplasty.

## Figures and Tables

**Figure 1 medicina-59-01608-f001:**
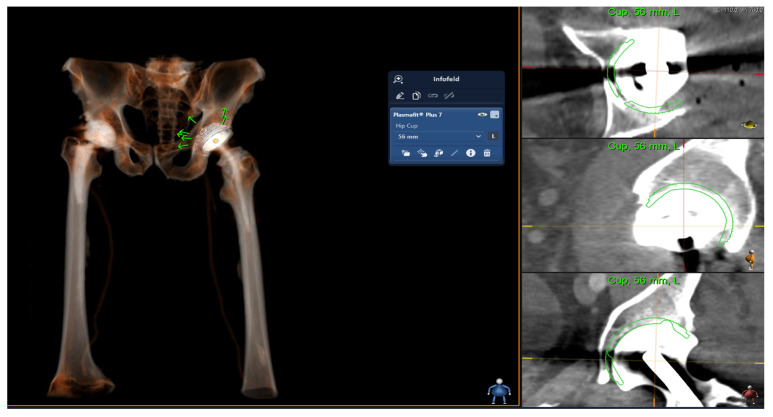
Preoperative planning of the acetabular cup (Plasmafit Plus 7) with 3D Joint Sectra (Sectra AB, Linköping, Sweden). The right side shows the actual planning based on CT imaging. Here, a hip joint spacer is currently still in place. The 56 mm size is shown to fit in all 3 planes (axial, sagittal and coronary) and was also confirmed intraoperatively.

**Figure 2 medicina-59-01608-f002:**
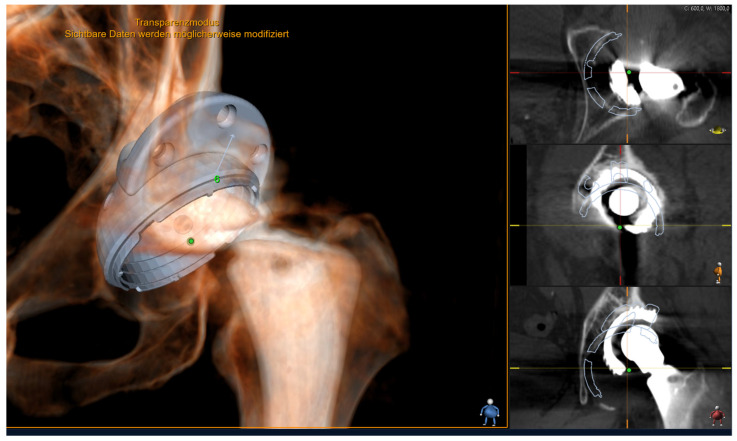
Personalized 3D templating using 3D Joint Sectra (Sectra AB, Linköping, Sweden) planning software. This image shows the preoperative planning of a Tritanium revision acetabular cup (size 66 mm) with a 62/20 mm acetabular augment. Stable fixation of the augment and the cup can be planned in three planes (shown on the right side of the picture).

**Table 1 medicina-59-01608-t001:** Acetabular cup revision.

Acetabular cup Revision	30
Aesculap Plasmafit - Plasmafit Plus - Plasmafit Plus 3 - Plasmafit Plus 7	10 2 4 4
Aesculap Plasmafit Revision Avantage Acetabular System	14 4
EcoFit 2M	1
Tritanium Revision	1

**Table 2 medicina-59-01608-t002:** Patients’ demographic data.

Age (mean ± SD)	71 ± 9.6
Female	14
Male	16
BMI (WHO classification in kg/m^2^)	
underweight (<18.5)	0
normal weight (18.5–24.9)	5
overweight (≥25.0)	15
obese (≥30.0)	10

SD standard deviation.

**Table 3 medicina-59-01608-t003:** Results of comparison 2D and 3D templating.

Accuracy of Templating		2D Templating	3D Templating	
Precise determination of acetabular cup (2D vs. 3D)		26.7% (8/30)	66.7% (20/30)	*p* < 0.002
Gender (2D vs. 3D)	Female Male	21.4% (3/14) 31.3% (5/16)	50% (7/14) 81.3% (13/16)	2D: *p* = 0.689 3D: *p* = 0.122
BMI (2D vs. 3D)	Normal weight Overweight Obese	0% (0/5) 46.7% (7/15) 10% (1/10)	60% (3/5) 73.3% (11/15) 60% (6/10)	2D: *p* = 0.765 3D: *p* = 0.812

## Data Availability

The data used to support the findings of the present study are available from the corresponding author upon request.

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
