# Peer review of "Accuracy of Personalized Computed Tomographic 3D Templating for Acetabular Cup Placement in Revision Arthroplasty"

_medicina, 2023, doi:10.3390/medicina59091608_

Round 1

Reviewer 1 Report

From the present manuscript, it should be noted that the predicted implant size generated through 3D templating tended to overestimate the implanted implant size by an average of 1.3 sizes. There were no statistically significant differences between the 2D and 3D templating methods regarding gender or BMI. Some comments need to addressed by authors as follows:

1.      Line 35, The novelty in the current article by the authors is too weak. The past has seen extensive published work of written material. It is required to provide more details for more explanation about the present novel in the introductory section.

2.      Line 36-37, related to hip arthroplasty explanation, please provide additional relevant reference as follows: https://doi.org/10.3390/ma14247554

3.      Line 45, it is the first time the authors mentioning “CT” I the main body manuscript. Before state it as CT, please state this extend as Computed tomography.

4.      Line 79, What is the baseline of patient selection? Is there any protocol, standard, or basis that has been followed? It is unclear since the patient is very heterogeneous with a small number. The resonance involved impacts the present result making this study flaws. One major reason for rejecting this paper.

5.      Line 91, the explanation of 2D and 3D would be encouraged to discuss.

6.      Line 111, the authors used “Joint Sectra” for Preoperative planning of the acetabular cup, but nothing any explanation regarding it in the materials and methods?

7.      Computed tomography has been widely adopted in medical investigation. Please provide this information along with relevant reference as follows: https://doi.org/10.3390/ma16093298 and https://doi.org/10.3390/biomedicines11020427

-

Author Response

Thank you very much for your comments and constructive suggestions. I greatly appreciate your insightful feedback and valuable suggestions. Your expertise in this field is truly impressive, and we feel privileged to have your guidance.

To 1) We agree with you that the novelty of the article needs to be highlighted more and the current literature needs to be described in the introduction. We have increased our attention to the current literature and the novelty of the current article in the introduction, and we have considered and mentioned the most currently important papers and reviews:

“In a systematic review by Bishi et al, it was demonstrated that the 3D template provides higher accuracy and reliability in predicting implant size in preoperative planning compared with the 2D template in primary arthroplasty [24]. However, in both clinical practice and current literature, 2D templating still endures as the standard technique [26][27]. Existing studies on preoperative planning in revision arthroplasty have predominantly centered around the application of 3D printing [28][29].”

To 2) The reference was added. Thank you for this reference.

To 3) Thank you for this correction.

To 4) Thank you very much for this constructive comment. We have described our methods in more detail and better presented the inclusion and exclusion criteria.  “Inclusion criteria were all patients who underwent revision hip arthroplasty in our institution with reimplantation of an acetabular cup between April 2019 and November 2022 and who had routine preoperative computed tomography of the pelvis. Patients who did not undergo acetabular cup revision or who did not have preoperative computed tomography were excluded. The included implants were: Aesculap Plasmafit, B.Braun, Melsungen, Germany; Aesculap Plasmafit Plus 3, B.Braun, Melsungen, Germany; Aesculap Plasmafit Plus 7, B.Braun, Melsungen, Germany; Aesculap Plasmafit Revision, B. Braun, Melsungen, Germany; Avantage Acetabular System, Zimmer Biomet, Warsaw, Indiana, USA; EcoFit 2M, Implantcast, Buxtehude, Germany; Tritanium Revision, Stryker, Mahwah, New Jersey, USA.”

To 5) Thank you for this comment. We have described the 2D and 3D planning in more detail.

To 6) we have added the term 3D Joint Sectra in our methods. This was not explained correctly before. Thank you very much for this comment.

To 7) The information and references have been included in the discussion. Thank you for this information.

I would like to express my sincere gratitude for your invaluable suggestions, which have significantly enhanced the quality of this manuscript.

Reviewer 2 Report

To date, there are no studies available regarding the accuracy of 3D templating in revision hip arthroplasty. The objective of this study is to assess the precision in acetabular cup positioning using computed tomographic (CT) 3D templating in revision arthroplasty compared to conventional 2D templating.

L65:  The following study is different from your study? what is different ?

          Edit introduction. And discuss it on the discussion section.

3D Printing Aids Acetabular Reconstruction in Complex Revision Hip Arthroplasty. Adv Orthop. 2017 https://pubmed.ncbi.nlm.nih.gov/28168060/

L70: Describe the diagnosis for revision arthroplasty.

         Infection? Aseptic looseing?

L116: Who is surgeon? Only One same expert surgeon? I think that surgeon level affects the accuracy.

L125: add the citation of Lewinnek safety zone.

L164: It is very unclear that primary outcome is the size. add the outcome on Lewinnek safety zone by postoperative Xp.

Results: Add the table 3 of main outcome and subgroup analysis on gender, BMI.

              Compare 2D and 3D.

L204: Discussion: 1st paragraph: write summary of results

Discussion: write your strength, clinical and research implication clearly.

L257: Too small sample is limitation.

Minor editing of English language required

Author Response

I extend my sincere appreciation for your insightful comments and invaluable suggestions. Your expertise in this field is truly commendable, and we hold it in high regard that our manuscript has been subjected to your expert review.

To L65) we have included the listed study in the introduction and discussion. Thank you for this important study, which we have discussed below in our discussion.

To L70) Thank you for this comment. We have added: “13 patients underwent surgery for aseptic loosening and 17 patients for infection.”

To L116) Thank you for this comment. We have added: “The surgery was performed by a one of three senior surgeons.“

To L125) we have added the citation of Lewinnek safe zone

To L167) Thank you very much for this constructive suggestion. We have added: “The acetabular cup alignment was within the Lewinnek safe zone in 93.3% (28/30) and had a mean inclination angle of 43.9°.”

Results: Presenting the main result and the subgroup analysis on gender and BMI using a table leads to a clearer presentation of our results. Thank you very much for this suggestion.              

Table 3. Results of 2D and 3D templating

2D templating

3D templating

Accuracy of templating

26.7% (20/30)

66.7% (8/30)

p<0.002

Gender

Female

Male

21.4% (3/14)

31.3% (7/14)

50% (5/16)

81.3% (13/16)

2D: p=0.689

3D: p=0.122

BMI

Normal weight

Overweight

Obese

0% (0/5)

46.7% (7/15)

10% (1/10)

60% (3/5)

73.3% (11/15)

60% (6/10)

2D: p=0.765

3D: p=0.812

To L204) We summarized our findings in the discussion: “The main finding of our study is that 3D templating based on computed tomography leads to an increased accuracy of predicting the size of acetabular cups in revision arthroplasty and therefore provides a reliable tool for preoperative planning in revision hip arthroplasty.“

Discussion: The discussion has been revised.

To L257) we have added as Limitation: “Second, we studied only a small cohort of patients.”

Reviewer 3 Report

The authors need to address the following comments before accepting for publication

1. All abbreviations need to be reported at the beginning or at the end of the manuscript. Eg:- three-dimensional (3D) , two-dimensional (2D)

2. The CT scans are also to be used to model and to do analysis for hip implants, this can be added in the introduction and also the authors can add the following supporting papers

Static structural analysis of the effect of change in femoral head sizes used in Total Hip Arthroplasty using finite element method

https://www.tandfonline.com/doi/full/10.1080/23311916.2022.2027080

Evolution of different designs and wear studies in total hip prosthesis using finite element analysis: A review

https://www.tandfonline.com/doi/full/10.1080/23311916.2022.2027081

Wear estimation of hip implants with varying chamfer geometry at the trunnion junction: a finite element analysis

https://pubmed.ncbi.nlm.nih.gov/36716460/

Wear estimation at the contact surfaces of oval-shaped hip implants using finite element analysis

https://www.tandfonline.com/doi/full/10.1080/23311916.2023.2222985

Limitations of the current work have to be reported 

Conclusions has to be reframed

Not required

Author Response

Thank you very much for your comments and valuable suggestions. You are so professional in this area and we are honored to have this manuscript reviewed by you.

To 1) The supporting papers were included in the introduction and greatly enhance the introduction of the manuscript.

To 2) The conclusion has been revised.

I wish to convey my appreciation for your invaluable recommendations, which have markedly elevated the caliber of this manuscript.

Round 2

Reviewer 2 Report

Thank you for revision.

Table 3: Add the results of comparison of  2D VS 3D. For example, 2D VS 3D in Female

Author Response

Thank you very much for your comments and constructive suggestions. We have revised table 3. Thank you for this correction!   Table 3. Results of comparison 2D and 3D templating

Accuracy of templating

2D templating

3D templating

Precise determination of acetabular cup

(2D vs 3D)

26.7% (20/30)

66.7% (8/30)

p<0.002

Gender

(2D vs 3D)

Female

Male

21.4% (3/14)

31.3% (7/14)

50% (5/16)

81.3% (13/16)

2D: p=0.689

3D: p=0.122

BMI

(2D vs 3D)

Normal weight

Overweight

Obese

0% (0/5)

46.7% (7/15)

10% (1/10)

60% (3/5)

73.3% (11/15)

60% (6/10)

2D: p=0.765

3D: p=0.812